# Subjective Oral Health-Related Quality of Life and Objective Oral Health in People with Ectodermal Dysplasia

**DOI:** 10.3390/ijerph18010143

**Published:** 2020-12-28

**Authors:** Nils Niekamp, Johannes Kleinheinz, Daniel R. Reissmann, Lauren Bohner, Marcel Hanisch

**Affiliations:** 1Department of Cranio-Maxillofacial Surgery, Research Unit Rare Diseases with Orofacial Manifestations, University Hospital Münster, Albert-Schweitzer-Campus 1, Building W 30, D-48149 Münster, Germany; n_niek01@uni-muenster.de (N.N.); johannes.kleinheinz@ukmuenster.de (J.K.); lauren.bohner@ukmuenster.de (L.B.); 2Department of Prosthetic Dentistry, Center for Dental and Oral Medicine, University Medical Center Hamburg-Eppendorf, Martinistrasse 52, D-20251 Hamburg, Germany; d.reissmann@uke.de

**Keywords:** rare diseases, ectodermal dysplasia, oral health, OHIP-14, EDS, patient-reported outcome, PhOX

## Abstract

Ectodermal dysplasia (ED) refers to a heterogeneous group of genetic diseases of the skin, skin appendages, and teeth. People with ED experience a poorer oral health-related quality of life (OHRQoL) compared to the general population. The aim of this study was to examine the OHRQoL of people with ED and to measure their objective physical oral health to confirm or disprove evidence of poorer oral health in this population. To determine OHRQoL, the German version of the 14-item Oral Health Impact Profile (OHIP-14G) was used. All the participants in the study were clinically examined, and the measured parameters were recorded using the Physical Oral Health Index (PhOX). In total, 10 male and 11 female participants, with an average age of 22.0 ± 9.0 years, were included in this study. The OHIP-14G summary score was 23.9 (±15.2) points (range: 0–56 points). The PhOX summary score was 61.2 (±5.1) points (range: 22–80 points). The findings indicated that both the OHRQoL and physical oral health of the participants were highly impaired and that their objective and subjective oral health were worse than those of the general population in Germany.

## 1. Introduction

Ectodermal dysplasia (ED) comprises a heterogeneous group of genetically determined diseases of the skin and skin appendages, with malformations of derivatives of the ectoderm. In addition to manifestations in hair, nails, sweat glands, sebaceous glands, mammary glands, and ciliary glands, it can also occur in areas relevant to the field of dentistry, in the form of tooth non-dispositions and form anomalies of the teeth. Oral manifestations of the various forms of ED not only include dental agenesis (e.g., hypodontia, oligodontia, and anodontia), shape anomalies of the teeth, such as microdontia and conical teeth, and reduced salivary flow rates, but also forms such as cleft formation, including ectrodactyly ectodermal dysplasia cleft lip-palate syndrome [1,2,3,4,5,6,7]. In addition to tooth development, mineralization can be disturbed, and tooth eruption can be problematic. This may reduce a person‘s ability to chew and speak [8], which can lead to psychological problems, particularly in those affected by oligodontia [9]. About one in 5000–10,000 people is affected by ED [10], and almost 200 clinically or genetically distinct forms of ED have been recorded. A new classification for ED was recently published, which classifies and organizes ED according to phenotype, genotype, and molecular signaling pathway (Table 1) [2]. Despite the obvious symptoms, such as multiple missing teeth, there are frequently problems in diagnosing some forms of ED [11,12].

Oral manifestations are also associated with reduced oral health-related quality of life (OHRQoL) among people with ED. In a study conducted by our working group, the 14-item Oral Health Impact Profile (OHIP-14) score identified an inferior OHRQoL for people with ED, compared to the general population, in Germany [12]. However, the study findings indicated that previous research had only addressed the perceptions of OHRQoL from the perspective of study participants. The aim of this study was therefore to examine the OHRQoL of people with ED and to measure their objective physical oral health to confirm or disprove evidence of poorer oral health in this population.

## 2. Methods

The data were collected between August 2019 and March 2020. Ethical approval for the study was obtained from the Medical Association of Westphalia-Lippe and the Westphalian Wilhelm University of Münster (No. 2019-402-f-S).

### 2.1. Inclusion Criteria

The participants had to be at least 18 years old and affected by ED.

### 2.2. Participants

People with ED were asked by the Selbsthilfegruppe Ektodermale Dysplasie e.V. (Germany) to participate in the study during the special consultation hour “Rare diseases with oral involvement”.

### 2.3. Assessment of Physical Oral Health

The Physical Oral Health Index (PhOX) [13] was developed for the purpose of recording and quantifying all aspects of the physical oral health of subjects. It consists of a self-assessment, as well as extra- and intraoral findings. These categories are further divided into five subcategories with 14 items in total (see Appendix A and Table 2).

Each criterion is evaluated on a five-point ordinal rating scale ranging from 0 to 4 and, depending on relevance, weighted either one, two, or three times (see Table 2). This yields a total score ranging from 0 to 100 points, where 0 points is the worst possible physical oral health and 100 points the best. The periodontal status of the participants in this study was ascertained with the help of Ramfjord’s Periodontal Disease Index [14] on teeth 16, 21, 24, 36, 41, and 44.

All the PhOX physical examinations were performed by the same dentist with 10 years of professional experience at a university hospital and the same student. The data were noted on the questionnaire and then transferred to an Excel spreadsheet.

### 2.4. Assessment of OHRQoL (OHIP)

To determine OHRQoL, the OHIP-14G [15] was used (see Appendix A), which is the German 14-item short version of the OHIP-49. The OHIP-14G consists of 14 items to assess the frequency of pain, restrictions, social, or physical stress, discomfort, and difficulties relating to social life. These items are rated on a scale of 0–4, where indicates 0 “never”, and 4 “very often”. The total score ranges from 0, meaning no negative impact, to 56, indicating a very high negative impact of oral health on quality of life. The OHIP-14G questionnaire was handed out to the participants to complete on their own.

### 2.5. Statistical Methods

The descriptive data were assessed using SPSS Statistics for Windows version 26.0 (IBM Corp., Armonk, NY, USA) and SAS software version 9.4 (SAS Institute Inc., Cary, NC, USA). The data were presented descriptively and graphically to illustrate the participants’ oral conditions and their perceptions of their oral health quality.

The general data (i.e., age, age at time of diagnosis, and time between first symptom and diagnosis) were described. The clinical conditions were indicated by the participants as nominal data (yes/no), qualitative data, or ordinal data, depending on the oral manifestation to be analyzed. The participants’ distribution regarding the different oral conditions were presented as percentage values. The OHIP-14G and PhOX values were presented as mean ± standard deviation, median, and 95% confidence interval. The minimal and maximal values were also reported.

## 3. Results

### 3.1. Participants

In total, 10 male and 11 female participants, with an average age of 22.0 ± 9.0 years, took part in this study. The time between the first symptom and diagnosis was 4.0 ± 23.0 years, whereas the age at the time of diagnosis was 13.0 ± 22.0 years.

Among the participants, 45.5% presented with between one and eight teeth, 31.8% presented with between nine and 16 teeth, and 22.7% had more than 16 teeth. “Filled” teeth were presented in 31.8% of the cases. Of these, three participants had one to two restorations, and four participants reported having between five and nine restorations. An enamel or tooth defect was observed in 26.3% of cases.

All the participants presented with an “involvement of the oral cavity”, and 14 had previously received orthodontic treatment. Regarding the sensation of dry mouth, two participants confirmed a sensation of dryness, 12 reported partial dryness, and seven had no such symptoms. Seven participants regarded their saliva consistency as fluid, and 15 noted that it was viscous. None of the participants reported a total absence of saliva.

### 3.2. Physical Oral Health

The total PhOX values were 61.20 ± 5.06 points (range: 22–80 points). The percentages for each item are presented in Table 3 and Figure 1.

### 3.3. OHRQoL

The OHIP-14G values are presented as percentages in Table 4 and Figure 2. The total OHIP-14G scores were 23.85 ± 15.17 within the range of the minimum (0) and maximum (56) values.

## 4. Discussion

This is the first study to investigate OHRQoL and physical oral health and their association in patients with ED. Even though only 21 participants could participate in the study, the findings indicated that both OHRQoL and physical oral health were highly impaired, and that the participants’ objective and subjective oral health were worse than those of the general population in Germany [13,16].

The questionnaires used in this study have not been validated for patients with rare diseases such as ED. This has both a positive and a negative side, as the data could be applicable to other types of rare diseases as long as they affect the oral cavity.

The average OHIP-14G value of the study participants was 23.85 ± 15.17. This value was higher than that of the average German population (4.09), indicating worse oral health, on average, among the study participants [13]. Higher OHIP-14G values were also measured in people with ED (12.23 men; 11.79 women), compared to the average German population in a previous study by our research group [9]. The results of that study were lower than those of the present study, even though the number of participants in this study was much higher at 110. The possibility of a direct comparison is therefore limited. Nevertheless, the reduced OHRQoL of people with ED appears to be further confirmed.

None of the study participants exhibited the minimum possible score of 0 points (no effect of oral health on quality of life), and none attained the maximum score of 56 points (very high effect of oral health on quality of life). Their oral health had the least impact on the participants’ OHRQoL in terms of “Be irritable with others”, “Difficulty doing usual jobs”, and “Difficulty pronouncing words”, while their oral health most affected “Uncomfortable to eat”, “Uncertainty”, and “Quality of Life”. The evaluation of the PhOX has previously demonstrated that patients with ED achieve a worse result than subjects not affected by ED [13].

This finding indicates that the participants in this study also measurably exhibited reduced OHRQoL. None of the study participants had the lowest PhOX score of 0 points (worst), nor did they reach the maximum score of 100 points (best). With an average value of 61.20 ± 5.06, the participants were at least in the middle-to-better range of the points scale. The highest (best) values were achieved in the categories “Continuity”, “Surface”, and “Endodontium”, while the lowest scores were achieved in the categories “Mouth opening”, “Teeth quantity”, and “Hard tooth substance”. Thus, the values determined by the PhOX suggested that a reduced number of teeth could be a factor in a reduced, measurable OHRQoL.

It was also notable that 12 participants showed partial dryness of the mouth, and two exhibited complete dryness of the mouth. The saliva consistency of 15 participants was slightly frothy or more viscous. The reduced salivary flow rate and its negative effect on the hard tooth substance (i.e., lack of remineralization, increased susceptibility to caries) could provide a possible explanation for the poor values in the “Hard tooth substance” category. Future investigations should therefore ideally aim to determine the extent to which changed saliva consistency, as well as (partial) dryness of the mouth, affect the remaining teeth in ED patients. It could be proven that the majority of ED patients have lost multiple teeth, in some cases, to a very high degree. For example, about half of the participants in this study had less than nine teeth.

In this study, the psychological stress of ED could be demonstrated to some extent: the comparatively high OHIP-14G values in the categories “Uncertainty” and “Quality of life” reflect their negative influence on OHRQoL. Consequently, it can be demonstrated, on the basis of the data available here, that the known reduced OHRQoL of people in ED can also be measurably proven. Accordingly, people with ED show both subjectively and measurably worse oral health than is typical for the general population.

It would be interesting to determine the OHRQoL of ED patients after they have undergone functional chewing treatment, for example, with dental implants. If this can lead to a significant improvement in OHRQoL, national health care systems should possibly grant appropriate access to these forms of care to those affected, depending on their performance. The main treatment option would be implant prosthetic restorations [8]. Studies have thus far indicated a good survival rate associated with implants in patients with ED: an average of 84.6% after 20 years [17]. However, implantological measures can be made more difficult by the reduced bone quality of ED patients [18,19]. As a result, augmentation measures, such as a bone graft from the iliac crest or the retromolar region, as well as sinus floor elevation, should be considered [20].

If the results of this study are placed in the context of the rather low average age of the participants (22 years), it becomes clear how important treatment concepts are for the dental rehabilitation of these patients. The possible psychological problems that can occur in connection with the absence of many teeth in ED patients should always be taken into account [9] and have also been described in other studies with rare diseases [21]. This requires additional support for affected patients, especially in the dental field, to improve their OHRQoL through functional chewing rehabilitation.

### Limitations

The sample enrolled in this study was not very large, but given the rarity of the disease, it can be considered sufficient. A further limitation of the study was the use of scales to measure the quality of life in rare diseases.

## 5. Conclusions

The findings indicated that both OHRQoL and physical oral health were highly impaired in the study population, and that their objective and subjective oral health were worse than those of the general population in Germany.

## Figures and Tables

**Figure 1 ijerph-18-00143-f001:**
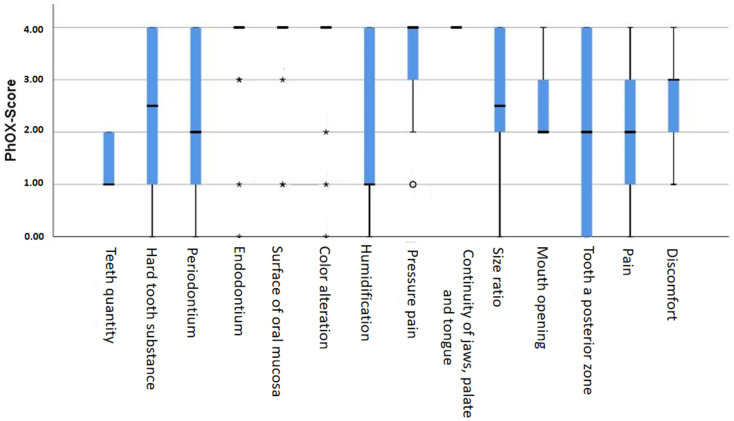
Box plot of PhOX item values. The missing plots at “Endodontium”, “Surface”, “Color alteration” and “Continuity” represent concentrations of samples with the same y-value (4.0). The asterisks and ball marks indicate outliers.

**Figure 2 ijerph-18-00143-f002:**
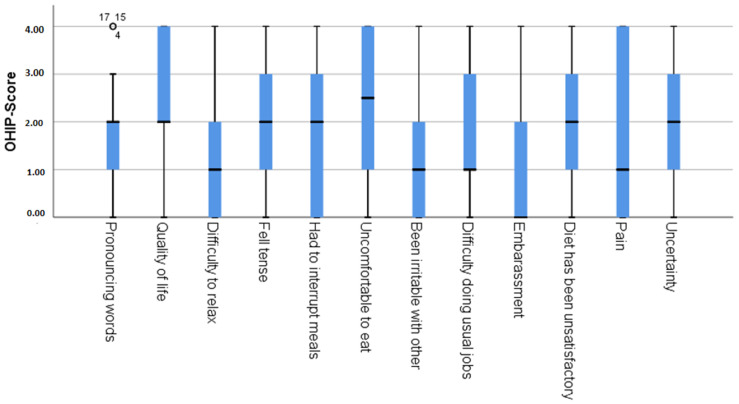
Box plot of the OHIP-14G item values.

**Table 1 ijerph-18-00143-t001:** Ectodermal dysplasia: Classification and organization by phenotype, genotype, and molecular signaling pathway (adapted from Wright et al. [2]).

Syndrome Name(s)	Gene(s)	Distinguishing Features
**Hypohidrotic ectodermal dysplasia (ED1; Christ–Siemens–Touraine syndrome)**	Ectodysplasin A, EDA (300451)	Hypohidrosis, hypotrichosis, hypodontia, formanomalia of the teeth, smooth dry skin, craniofacial dysmorphology, periorbital pigmentation
**Hypohidrotic ectodermal dysplasia 10A**	Ectodysplasin A receptor, EDAR (604095) or EDARADD (606603)	Hypohidrosis, hypotrichosis, hypodontia, smooth dry skin, craniofacial dysmorphology, periorbital pigmentation
**Hypohidrotic ectodermal dysplasia 10B**	Ectodysplasin A receptor, EDAR (604095) or EDARADD (606603)	Hypohidrosis, hypotrichosis, hypodontia, smooth dry skin, craniofacial dysmorphology, periorbital pigmentation
**Incontinentia pigmenti (IP)**	IKBKG (300248)	Short stature, cataract, microphthalmia, hypodontia, formanomalia of the teeth, extra ribs, breast aplasia, staged skin involvement, nail dystrophy, atrophic hair
**Ectodermal dysplasia and immunodeficiency 1 (EDAID1)**	IKBKG (300248)	Hypohidrosis, hypotrichosis, morbidity/mortality secondary to immunodeficiency
**Focal dermal hypoplasia (Goltz syndrome)**	PORCN (300651)	Short stature, facial asymmetry, narrow auditory canals, hearing loss, oral papillomas, hypodontia, syndactyly, sparse hair, skin atrophy
**Odontoonychodermal dysplasia (OODD)**	WNT10A (606268)	Sparse eyebrows, severe hypodontia, formanomalia of the teeth, smooth tongue, hyperhidrosis, hyperkeratosis, dystrophic nails, sparse eyebrows, thin hair
**Schopf–Schulz–Passarge syndrome**	WNT10A (606268)	Hypodontia, eyelid cysts, keratoderma, hypoplastic nails, hypotrichiosis
**Acro–dermato–ungual–lacrimal–tooth syndrome (ADULT syndrome)**	TP63 (603273)	Lacrimal obstruction, hypodontia, dysplastic teeth, breast hypoplasia, ectrodactyly, thin skin, dysplastic nails
**Ankyloblepharon-ectodermal defects-cleft lip and palate (AEC) syndrome (Hay–Wells syndrome)**	TP63 (603273)	Scalp erosions, conductive hearing loss, maxillary hypoplasia, lacrimal duct atresia, hypotrichosis, ankyloblepharon, cleft lip, hypodontia
**Rapp–Hodgkin syndrome**	TP63 (603273)	Short stature, maxillary hypoplasia, hearing loss, cleft lip and palate, hypodontia, syndactyly, thin skin, hypohidrosis
**Ectrodactyly, ectodermal dysplasia, and cleft lip and palate syndrome 3 (EEC3)**	TP63 (603273)	Blepharophimosis, cleft lip and palate, microdontia, hypodontia, syndactyly, hypokeratosis, dystrophic nails, hypotrichiosis
**Limb-mammary syndrome (LMS)**	TP63 (603273)	Lacrimal duct atresia, hypodontia, cleft lip and palate, hypoplastic breasts, syndactyly, ectrodactyly, nail dystrophia
**Ectodermal dysplasia, ectrodactyly, and macular dystrophy syndrome (EEMS)**	CDH3 Cadherin 3 (114021)	Sparse scalp hair, eyebrows and eyelashes, hypodontia, small teeth, ectrodactyly, syndactyly, camptodactyly, normal sweating
**Ectodermal dysplasia 4, hair/nail type (ECTD4)**	KRT85 Keratin 85 (602767)	Nail dystrophy, onycholysis, absent eyebrows/eyelashes, alopecia, normal skin/teeth
**Ectodermal dysplasia/skin fragility syndrome**	PKP 1 Plakophilin 1 (601975)	Nail dystrophy and thickening, hypotrichosis, sweat glands, skin fragility
**Monoilethrix (MNLIX)**	Keratins 81, 86, 83; KRT81, KRT86, KRT83 (602153, 601928, 602765)	Follicular keratosis, nail dystrophy, hypotrichosis, brittle hair
**Cleft lip/palate-ectodermal dysplasia (CLPED1)**	Nectin 1 NECTIN1 (600644)	Malar hypoplasia, hypotrichosis, cleft lip/palate, hypodontia, syndactyly, onychodysplasia
**Arthorgryposis and ectodermal dysplasia**	Unknown	Short stature, microcephaly, cataract, cleft lip and palate, oligodontia, enamel defects, arthrogryposis, hypohidrosis, onychodysplasia
**Dermoodontodysplasia**	Unknown	Trichodysplasia, onychodysplasia, dental anomalies

**Table 2 ijerph-18-00143-t002:** Physical Oral Health Index (PhOX) domains and weights with the range for the values of each item. Translated domains and items from the Physical Oral Health Index.

Domain	#	Item	Weight	Range
Teeth and surrounding tissue	1	Number of teeth	3	0–12
	2	Tooth structure	3	0–12
	3	Periodontium	3	0–12
	4	Endodontia	2	0–8
Soft tissue intraoral	5	Surface	1	0–4
	6	Color	2	0–8
	7	Moisturization	1	0–4
Soft tissue and jaw	8	Pain on palpation	1	0–4
	9	Continuity	1	0–4
	10	Proportion	1	0–4
Function	11	Mouth opening	1	0–4
	12	Supporting area	3	0–12
Perception	13	Pain	2	0–8
	14	Paresthesia	1	0–4

**Table 3 ijerph-18-00143-t003:** Total PhOX values per item given by the participants (expressed as percentages).

PhOX Values (%)					
Domain	0	1	2	3	4
Teeth quantity	0	68.2	31.8	0	0
Condition of teeth	18.2	9.1	22.7	9.1	40.9
Condition of periodontium	4.5	22.7	45.5	0	27.3
Condition of endodontium	4.5	4.5	0	9.1	81.8
Surface of oral mucosa	0	9.1	0	4.5	86.4
Color of oral mucosa	4.5	4.5	4.5	0	86.4
Moistening of oral mucosa	9.1	59.1	0	0	31.8
Pain on palpation of jaws and muscles	0	9.1	4.5	13.6	63.6
Continuity of jaws, palate and tongue	0	0	0	0	100
Size ratio of jaws	4.5	18.2	27.3	13.6	36.4
Mouth opening capacity	0	0	72.7	9.1	18.2
Number of supporting zones	31.8	4.5	27.3	4.5	31.8
Pain frequency	9.1	22.7	2.3	27.3	9.1
Paresthesia frequency	0	13.6	22.7	36.4	22.7

**Table 4 ijerph-18-00143-t004:** Total 14-item Oral Health Impact Profile (OHIP-14G) values per item given by the participants (expressed as percentages), where 0 = never; 1 = hardly ever; 2 = occasionally; 3 = often; 4 = very often. The results were obtained from each question of the OHIP-14G questionnaire.

OHIP-14G Values (%)					
Domain	0	1	2	3	4
Difficulty pronouncing words	18.2	27.3	31.8	9.1	13.6
Taste	59.1	27.3	0	0	13.6
Quality of life	9.1	9.1	36.4	13.6	31.8
Difficulty relaxing	27.3	27.3	22.7	0	22.7
Felt tense	13.6	27.3	22.7	18.2	18.2
Had to interrupt meals	31.8	13.6	22.7	9.1	22.7
Uncomfortable to eat	13.6	13.6	22.7	18.2	31.8
Been irritable with others	36.4	18.2	18.2	13.6	9.1
Difficulty doing usual jobs	22.7	36.4	9.1	13.6	18.2
Unable to function	59.1	13.6	9.1	4.5	13.6
Embarrassment	22.7	22.7	9.1	22.7	22.7
Diet has been unsatisfactory	31.8	22.7	9.1	9.1	27.3
Pain	18.2	18.2	31.8	13.6	18.2
Uncertainty	9.1	18.2	27.3	18.2	27.3

## Data Availability

The datasets supporting the conclusions of this article are available from the Department of Cranio-Maxillofacial Surgery, University Hospital Münster, Germany.

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
