# Peer review of "Subjective Oral Health-Related Quality of Life and Objective Oral Health in People with Ectodermal Dysplasia"

_ijerph, 2020, doi:10.3390/ijerph18010143_

Round 1

Reviewer 1 Report

The authors have conducted a descriptive study to understand the subjective oral health-related quality of life (OHRQoL) as well as objective oral health clinical parameters in patients with ectodermal dysplasia (ED). For the OHRQoL, they utilized the German version of the 14-item Oral Health Impact Profile (OHIP-14) instrument. For the clinical parameters, the authors utilized the Physical Oral Health Index (PhOX). Overall, the study is conducted and presented well. Please consider the following suggestions which could potentially improve the manuscript.

  1. Figure 1 appears to be directly copied from reference 2. Please obtain necessary permission to reproduce the same. If not, please create original figure or table. This information is better presented as a table.
  2. The supplemental information is in German. Please provide English translation for English language readers.
  3. The assessment using PhOX (page 4, section 2.3) should be expanded to include the steps in detail on how the clinical examination was done and how the data was recorded.
  4. Figure 2 – The box plot is not easy to read with error bars being cut off. Also, some of the terms are not in modern English – parodontium. Periodontium is better known. Certain terms like ‘continuity’, ‘surface’, needs further explanation in the text. Please explain the values with asterisk symbol.
  5. PhOX instrument items should be explained further in the text. For example, what is being measured in the ‘continuity of jaws, palate, tongue’? How does this affect the OHRQoL?
  6. Limitations: Besides the low sample size, the subjective data obtained from patients is prone to bias.

Author Response

We would like to thank the editor and the reviewers for their time spent on reviewing our manuscript and their helpful comments. Their suggestions have been implemented in the manuscript. In this letter, we respond point-by-point to the comments and explain the revisions.

All changes to the manuscript were highlighted using the "Track Changes" function in Microsoft Word.

We hope the manuscript is now suitable for publication in the International Journal of Environmental Research and Public Health.

Reviewer 1:

The authors have conducted a descriptive study to understand the subjective oral health-related quality of life (OHRQoL) as well as objective oral health clinical parameters in patients with ectodermal dysplasia (ED). For the OHRQoL, they utilized the German version of the 14-item Oral Health Impact Profile (OHIP-14) instrument. For the clinical parameters, the authors utilized the Physical Oral Health Index (PhOX). Overall, the study is conducted and presented well. Please consider the following suggestions which could potentially improve the manuscript.

1.Figure 1 appears to be directly copied from reference 2. Please obtain necessary permission to reproduce the same. If not, please create original figure or table. This information is better presented as a table.Answer: thanks for this note, we have now created a table adapted to Wright et al.  

2. The supplemental information is in German. Please provide English translation for English language readers.Answer: we added English versions to the supplements

3. The assessment using PhOX (page 4, section 2.3) should be expanded to include the steps in detail on how the clinical examination was done and how the data was recorded.Answer: we have expanded the explanation of the PhOx under 2.3 and included a table. Also we added information on data recording:

“The PhOX was developed for the purpose of recording and quantifying all aspects of the physical oral health of subjects. It consists of the self-assessment, extra-oral findings and intra-oral findings. These categories are further subdivided into 5 sub-categories with altogether 14 items (see S1 and table 1).

Each criterion is evaluated on a 5-point ordinal rating scale ranging from 0 to 4 and, depending on relevance, weighted either one, two or three times (see table 1). This yields a total score of 0 up to 100 points, whereby 0 points is the worst possible and 100 points the best possible physical oral health. The periodontal status of the participants was ascertained with the help of Ramfjord’s Periodontal Disease Index (PDI)[16] on teeth 16, 21, 24, 36, 41 and 44.

Table 2. PhOX domains and weights with the range for the values of each item. Translated domains and items from the Physical Oral Health Index.

Domain

#

Item

Weight

Range

Teeth and surrounding tissue

1

number of teeth

3

0-12

2

tooth structure

3

0-12

3

periodontium

3

0-12

4

endodontia

2

0-8

Soft tissue intraoral

5

surface

1

0-4

6

color

2

0-8

7

moisturization

1

0-4

Soft tissue and jaw

8

pain on palpation

1

0-4

9

continuity

1

0-4

10

proportion

1

0-4

Function

11

mouth opening

1

0-4

12

supporting area

3

0-12

Perception

13

pain

2

0-8

14

paresthesia

1

0-4

The examination was always performed by a dentist with 10 years of professional experience at an University Hospital together with the same student. The data was noted on the questionnaire and then transferred to an Excel spreadsheet.”

4. Figure 2 – The box plot is not easy to read with error bars being cut off. Also, some of the terms are not in modern English – parodontium. Periodontium is better known. Certain terms like ‘continuity’, ‘surface’, needs further explanation in the text. Please explain the values with asterisk symbol.-> We improved the graph to make it clearly. We also changed the terms (i.e. “Periodontium” instead “Parodontium”; “Continuity of jaws, palate and tongue” instead “Continuity”; “Surface of oral mucosa” instead “Surface”). Explanation about missing plots and asterisks were included on the legend.

5. PhOX instrument items should be explained further in the text. For example, what is being measured in the ‘continuity of jaws, palate, tongue’? How does this affect the OHRQoL?Answer: we have now further explained the PhOX instrument and added a table, see above

6. Limitations: Besides the low sample size, the subjective data obtained from patients is prone to bias.Answer: This is absolutely correct. Our aim was therefore to collect objective data in addition to subjective data in order to provide an indication of the reduced oral health in people with ectodermal dysplasia  

Reviewer 2 Report

Thank you for the opportunity to review this manuscript, that investigates the oral health related quality of life and physical oral health of people with ectodermal dysplasia. I strongly support the need to build better evidence about the issues, particularly subjective impacts, caused by conditions like ED and so thank the authors for investigating this significant condition. However, I think the overall merit of the publication is limited by the small sample size, poor reporting of study methods and overall failure to discuss and acknowledge key limitations.

The aim for example, is not clear - was it to investigate oral health or oral health related quality of life, or both? The methods section is grossly lacking details, for example, about the consultations: where were these undertaken? At a dental clinic or another setting? By whom? Dentists or someone else? Can you please provide references for the two main indices in the methods. Can you also please clarify whether there was one or more examiner/assessor and whether they were trained and calibrated? Can you also please make it very clear that only self-reported data was used to determine physical oral health? It is unclear whether patients were physically examined. This is especially the case for saliva consistency, which is not mentioned at all until the results. Even regarding 'general data' please be specific about the type of data collected and from where. 

In the discussion, the authors report they have conducted a much larger study of OHRQoL in adults with ED. Therefore how can this much smaller, underpowered study be justified? What makes it novel or important? In addition, while the authors do acknowledge the small sample size, they fail to acknowledge the important of several other major sources of bias, such as lack of a control group.

One final comment is the quality of English, which needs improvement for the reader to understand the meaning of the sentence.

Author Response

We would like to thank the editor and the reviewers for their time spent on reviewing our manuscript and their helpful comments. Their suggestions have been implemented in the manuscript. In this letter, we respond point-by-point to the comments and explain the revisions.

All changes to the manuscript were highlighted using the "Track Changes" function in Microsoft Word.

We hope the manuscript is now suitable for publication in the International Journal of Environmental Research and Public Health.

Reviewer 2:

Thank you for the opportunity to review this manuscript, that investigates the oral health related quality of life and physical oral health of people with ectodermal dysplasia. I strongly support the need to build better evidence about the issues, particularly subjective impacts, caused by conditions like ED and so thank the authors for investigating this significant condition. However, I think the overall merit of the publication is limited by the small sample size, poor reporting of study methods and overall failure to discuss and acknowledge key limitations.

1. The aim for example, is not clear - was it to investigate oral health or oral health related quality of life, or both?

Answer: The aim of this study was to examine OHRQoL and to measure objective physical oral health in people with ectodermal dysplasia to maintain or rebut evidence of poorer oral health in this population.

Despite the small number of 21 patients we could at least provide the indication that both OHRQoL and physical oral health are highly impaired. But of course these indications must be confirmed by future studies with higher numbers of participants

2. The methods section is grossly lacking details, for example, about the consultations: where were these undertaken? At a dental clinic or another setting? By whom? Dentists or someone else?

Answer: we now added this missing information in the text:All the PhOX physical examinations were performed by the same dentist with 10 years of professional experience at a university hospital and the same student. The data were noted on the questionnaire and then transferred to an Excel spreadsheet.” (Line 94-96)

3. Can you please provide references for the two main indices in the methods.

Answer: We added references (Reference 13 and15) for the two main indices in the methods.

4. Can you also please clarify whether there was one or more examiner/assessor and whether they were trained and calibrated?

Answer: we now added this missing information in the text:All the PhOX physical examinations were performed by the same dentist with 10 years of professional experience at a university hospital and the same student. The data were noted on the questionnaire and then transferred to an Excel spreadsheet.” (Line 94-96)

One of the authors (DR) is the developer of the PhOX, he did the calibration of NN and MH

5. Can you also please make it very clear that only self-reported data was used to determine physical oral health? It is unclear whether patients were physically examined. This is especially the case for saliva consistency, which is not mentioned at all until the results. Even regarding 'general data' please be specific about the type of data collected and from where. 

Answer: All participants were physical examinated (=PhOX) by a dentist with 10 years of professional experience at an University Hospital together with the same student. The OHIP-14 questionnaire on the other hand was handed out to the participants to fill out themselves. We added this information (line 94-103)

6. In the discussion, the authors report they have conducted a much larger study of OHRQoL in adults with ED. Therefore how can this much smaller, underpowered study be justified?

Answer: in our study at that time only the subjective oral health-related quality of life was surveyed with the ohip questionnaire [Hanisch, M.; Sielker, S.; Jung, S.; Kleinheinz, J.; Bohner, L. Self-Assessment of Oral Health-Related Quality of Life in People with Ectodermal Dysplasia in Germany. Int J Environ Res Public Health. 2019 May 31;16(11).]. The participants were not clinically and objectively examined at that time. With the present study we can now provide evidence that people with ectodermal dysplasia are both subjectively and objectively affected by reduced oral health

7. What makes it novel or important? In addition, while the authors do acknowledge the small sample size, they fail to acknowledge the important of several other major sources of bias, such as lack of a control group.

Answer: With the present study we can now provide evidence that people with ectodermal dysplasia are both subjectively and objectively affected by reduced oral health. Since we know the scores for both OHIP-14 and PhOX for the normal population, we can indicate that the oral health values for ectodermal dysplasia are worse than in the normal population

8. One final comment is the quality of English, which needs improvement for the reader to understand the meaning of the sentence

Answer: we have now had the manuscript linguistically revised by a native speaker

Reviewer 3 Report

The paper is an original study, investigating the Oral health related quality of life and

 Objective physical Oral Health in a population affected by Ectodermal dysplasia.

The topic is of current interest, since health related quality of life is deeply investigated nowadays and a great attention is posed to its use in cost-effectiveness analysis for decision making in health economy. Oral health needs to be considered as well.

The article is well written, nevertheless, some major criticisms should be highlighted:

An important limitation of the study is the absence of the control group.  Assessing the OHRQoL in a group of healthy subjects with similar characteristics to the study group is essential to draw conclusions.

You refer to the ref 13 for comparison (in German), but does it consider a similar population? I mean, regarding age and gender distribution etc.? Does it use a similar method of evaluation?

The sample enrolled in the study is not very large, but given the rarity of the disease, it can be considered sufficient. It has been addresses also in Limitations section.

In Table 2: I don’t understand the origin of Raw “Quality of life”. How was it determined? What question does it correspond to in the questionnaire?

It would be useful to provide an English version of the questionnaires.

Author Response

We would like to thank the editor and the reviewers for their time spent on reviewing our manuscript and their helpful comments. Their suggestions have been implemented in the manuscript. In this letter, we respond point-by-point to the comments and explain the revisions.

All changes to the manuscript were highlighted using the "Track Changes" function in Microsoft Word.

We hope the manuscript is now suitable for publication in the International Journal of Environmental Research and Public Health.

Reviewer 3:

The paper is an original study, investigating the Oral health related quality of life and

 Objective physical Oral Health in a population affected by Ectodermal dysplasia.

The topic is of current interest, since health related quality of life is deeply investigated nowadays and a great attention is posed to its use in cost-effectiveness analysis for decision making in health economy. Oral health needs to be considered as well.

The article is well written, nevertheless, some major criticisms should be highlighted:

1. An important limitation of the study is the absence of the control group.  Assessing the OHRQoL in a group of healthy subjects with similar characteristics to the study group is essential to draw conclusions.

Answer: you are absolutely right, a control group is missing. But since we know the scores for both OHIP-14 and PhOX for the normal population, we can indicate that the oral health values for ectodermal dysplasia are worse than in the normal population

2. You refer to the ref 13 for comparison (in German), but does it consider a similar population? I mean, regarding age and gender distribution etc.? Does it use a similar method of evaluation?

Answer: the reference is a specifically selected representative sample of the german norm population. Our aim was to compare our participants with the values known from these reference 13 (now reference 16).

3. The sample enrolled in the study is not very large, but given the rarity of the disease, it can be considered sufficient. It has been addresses also in Limitations section.

Answer: we now revised the limitations section (4.1.): “The sample enrolled in this study was not very large, but given the rarity of the disease, it can be considered sufficient. A further limitation of the study was the use of scales to measure the quality of life in rare diseases.” (Line 210-212)

4. In Table 2: I don’t understand the origin of Raw “Quality of life”. How was it determined? What question does it correspond to in the questionnaire?

Answer: Table 2 (now Table 4) refers to the results obtained from each question of the OHIP-14 questionnaire (see Supplement 2). We added this information to table 4. Line 142-144.

5. It would be useful to provide an English version of the questionnaires

Answer: we added English versions of the questionnaires to the supplements

Reviewer 4 Report

The following article brings novelty to science. The methodology used is correct. However, I would like to make the following recommendations

- The summary should not contain any "background, etc." sections. Review articles previously published by the journals.

- On line 50 correct writing errors and punctuation marks.

- On line 58 you should add at the end of the sentence of the new classification "figure 1".

- On line 92 correct "negative".

- The OHIP 14, has 14 items. In table 2 there are only 13 items, check it.

- In the discussion it should be added that the questionnaires used have not been validated in patients with rare diseases such as ectodermal dysplasia. This has a positive and a negative side as the data can be applicable to other types of rare diseases as long as they affect the oral cavity.

- The limitations of the study include not only the sample size but also the use of scales to measure quality of life in rare diseases.

Author Response

We would like to thank the editor and the reviewers for their time spent on reviewing our manuscript and their helpful comments. Their suggestions have been implemented in the manuscript. In this letter, we respond point-by-point to the comments and explain the revisions.

All changes to the manuscript were highlighted using the "Track Changes" function in Microsoft Word.

We hope the manuscript is now suitable for publication in the International Journal of Environmental Research and Public Health.

Reviewer 4:

The following article brings novelty to science. The methodology used is correct. However, I would like to make the following recommendations

1. - The summary should not contain any "background, etc." sections. Review articles previously published by the journals.

Answer: we removed “background, etc.” in the summary. We added this reference:

Oelerich, O.; Kleinheinz, J.; Reissmann, D.R.; Köppe, J.; Hanisch, M. Correlation between Oral Health-Related

Quality of Life and Objectively Measured Oral Health in People with Ehlers–Danlos Syndromes. International

Journal of Environmental Research and Public Health 2020, 17, 8243, doi:10.3390/ijerph17218243.

2. - On line 50 correct writing errors and punctuation marks.

Answer: we corrected writing errors and punctuation marks (line 50)

3. - On line 58 you should add at the end of the sentence of the new classification "figure 1".

Answer: we have now created a table (Table 1) and added at the end of the sentence “Table 1”. Line 56

4. - On line 92 correct "negative".

Answer: we corrected "negative" Line 102.

5. - The OHIP 14, has 14 items. In table 2 there are only 13 items, check it.

Answer: We included the missing item (Taste) on Table 2.

6. - In the discussion it should be added that the questionnaires used have not been validated in patients with rare diseases such as ectodermal dysplasia. This has a positive and a negative side as the data can be applicable to other types of rare diseases as long as they affect the oral cavity.

Answer: we added this sentence to the discussion section (line 153-155).

7. - The limitations of the study include not only the sample size but also the use of scales to measure quality of life in rare diseases.

Answer: we added this sentence to the limitations section (line 210-212).

Round 2

Reviewer 3 Report

The authors performed suggested modifications.